# Fine-tuning citrate synthase flux potentiates and refines metabolic innovation in the Lenski evolution experiment

Erik M Quandt[1,2], Jimmy Gollihar[1], Zachary D Blount[2,3], Andrew D Ellington[1,2,4,5], George Georgiou[1,4,6,7], Jeffrey E Barrick[1,2,4,5,8]*

[1]Institute for Cellular and Molecular Biology, The University of Texas at Austin, Austin, United States; [2]BEACON Center for the Study of Evolution in Action, Michigan State University, East Lansing, United States; [3]Department of Microbiology and Molecular Genetics, Michigan State University, East Lansing, United States; [4]Department of Molecular Biosciences, The University of Texas at Austin, Austin, United States; [5]Center for Systems and Synthetic Biology, The University of Texas at Austin, Austin, United States; [6]Department of Chemical Engineering, The University of Texas at Austin, Austin, United States; [7]Department of Biomedical Engineering, The University of Texas at Austin, Austin, United States; [8]Center for Computational Biology and Bioinformatics, The University of Texas at Austin, Austin, United States

*For correspondence: jbarrick@cm.utexas.edu

Competing interests: The authors declare that no competing interests exist

**Abstract** Evolutionary innovations that enable organisms to colonize new ecological niches are rare compared to gradual evolutionary changes in existing traits. We discovered that key mutations in the *gltA* gene, which encodes citrate synthase (CS), occurred both before and after *Escherichia coli* gained the ability to grow aerobically on citrate (Cit$^+$ phenotype) during the Lenski long-term evolution experiment. The first *gltA* mutation, which increases CS activity by disrupting NADH-inhibition of this enzyme, is beneficial for growth on the acetate and contributed to preserving the rudimentary Cit$^+$ trait from extinction when it first evolved. However, after Cit$^+$ was refined by further mutations, this potentiating *gltA* mutation became deleterious to fitness. A second wave of beneficial *gltA* mutations then evolved that reduced CS activity to below the ancestral level. Thus, dynamic reorganization of central metabolism made colonizing this new nutrient niche contingent on both co-opting and overcoming a history of prior adaptation.

## Introduction

Evolutionary descent with modification has produced organisms with complex metabolic and gene regulatory networks. When a microbial population encounters a new environment, mutants encoding more effective variants of these networks that improve nutrient utilization or reveal latent metabolic capabilities may evolve (*Ryall et al., 2012*). While there are many degrees of freedom that evolution can potentially access in altering cellular networks, only those mutational pathways that do not include deleterious intermediate steps are likely to be realized in most populations (*Bridgham et al., 2006*, *2009*; *Weinreich et al., 2006*). Laboratory studies have characterized mutational pathways that enable microorganisms to access new enzyme activities when there is strong selection for a new trait (*Hall, 2003*; *Näsvall et al., 2012*). Combinations of metabolic and regulatory mutations that enable microorganisms to evolve toward optimal growth rates under defined conditions have also

**eLife digest** Bacteria and other organisms are constantly under pressure to survive in the face of ever-changing environmental challenges. They generally adapt to these challenges through genetic mutations that modify features they already have. However, occasionally a species may acquire an entirely new characteristic – known as an evolutionary innovation – that allows it to colonize a new environment or adopt a new mode of life.

The Lenski Experiment, which began in 1988, is an ongoing study of the evolution of bacteria grown in the laboratory. The experiment started with twelve identical populations of bacteria and has so far tracked the genetic mutations that have been acquired by the populations over tens of thousands of generations. Fifteen years into the experiment, bacteria in one of the populations evolved the ability to exploit a new food source, a molecule called citrate. The bacteria in this population have multiple mutations in a gene called *gltA*, which encodes an enzyme called citrate synthase. However, it was not clear how these mutations contributed to the ability of the bacteria in this population to use citrate.

Here, Quandt et al. have used a variety of genetic and biochemical techniques to examine the mutations in *gltA*. They found that one mutation occurred before the bacteria evolved the ability to use citrate, and others occurred afterward. The first mutation in *gltA* increased the activity of the citrate synthase enzyme, which paved the way for a key mutation affecting citrate transport into cells that allowed the bacteria to consume the new food source. However, once the bacteria evolved the ability to use citrate, and more mutations in other genes refined this process, the increased citrate synthase activity became detrimental. At this point, the bacteria acquired a second *gltA* mutation that lowered citrate synthase activity to a level below what it had been in the original bacteria before the first *gltA* mutation.

The Lenski Experiment presents a rare opportunity to examine the complete history of an evolutionary innovation. Quandt et al. findings show that evolutionary 'reversals' may be necessary to adjust cell processes in different ways as an innovation first evolves and is further refined. A challenge for future work is to identify the other mutations that, together with the first *gltA* mutation, were necessary for the bacteria to evolve the ability to use citrate.

been exhaustively mapped at a whole-genome level (*Conrad et al., 2011*; *Tenaillon et al., 2012*). To this point, however, we have rarely had the opportunity to observe the interplay of these two regimes of optimization and innovation as they must occur over longer evolutionary timescales in nature (*Barrick and Lenski, 2013*).

The Lenski long-term evolution experiment (LTEE) with *E. coli* provides a unique opportunity to study how metabolic and regulatory networks have changed over a history spanning more than 25 years of microbial adaptation (*Lenski and Travisano, 1994*; *Lenski et al., 1991*). In particular, cells in one of the twelve LTEE populations evolved a qualitatively new metabolic capability—aerobic citrate utilization (Cit$^+$ phenotype)—that enabled them to colonize a previously unoccupied ecological niche (*Blount et al., 2008*). This Cit$^+$ innovation is highly beneficial because it grants access to an abundant and previously untapped nutrient. Yet, it is also very rare. So far, a Cit$^+$ variant has evolved in just one of the twelve LTEE populations, and then only after ~15 years of evolution. The rarity of the Cit$^+$ innovation suggests that accessing this new metabolic trait is contingent on a multi-step mutational pathway.

The evolution of aerobic citrate utilization in the LTEE involved three stages: potentiation, actualization, and refinement. Actualization refers to the first appearance of phenotypically Cit$^+$ cells. This transition was caused by a duplication that activated expression of the *citT* citrate:succinate antiporter gene through promoter capture (*Blount et al., 2012*). However, on its own this mutation confers only extremely weak citrate utilization. Subsequently, this rudimentary Cit$^+$ trait was refined to a stronger phenotype, Cit$^{++}$, when cells that were capable of fully exploiting citrate during each 24 hr growth cycle evolved, coincident with a large increase in cell density in this LTEE population (*Blount et al., 2008*). Chief among the refining mutations was a promoter mutation that activated expression of the *dctA* C$_4$-dicarboxylate:H$^+$ symporter gene (*Quandt et al., 2014*). Strains reconstructed with just the *citT* duplication and this *dctA** mutation are capable of

fully utilizing citrate (i.e., they are Cit$^{++}$) . However, this simple two-step mutational pathway was apparently inaccessible without the prior evolution of one or more unknown mutations that created a potentiated genetic background (*Blount et al., 2008*).

A whole-genome phylogeny of this LTEE population has provided candidates for other mutations that contributed to the potentiation and refinement steps of Cit$^{++}$ evolution (*Blount et al., 2012*). We show here that one target of interest is the *gltA* gene, which encodes the enzyme citrate synthase (CS). CS catalyzes the first irreversible step in the tricarboxylic acid (TCA) cycle: the aldol condensation of oxaloacetate (OAA) and acetyl-CoA to form citrate. Multiple mutations in one gene are rare in LTEE lineages that have retained the low ancestral mutation rate (*Barrick et al., 2009*; *Wielgoss et al., 2011*), yet the *gltA* gene was mutated twice in most Cit$^{++}$ isolates, once before and once after Cit$^{+}$ evolved. Due to the conspicuous metabolic function of CS and the appearance of the later *gltA* mutations specifically in Cit$^{+}$ isolates, it was previously hypothesized that these mutations refined the Cit$^{+}$ phenotype (*Blount et al., 2012*).

In this study, we characterized the effects of *gltA* mutations on competitive fitness, mRNA expression levels, and enzyme activity. Integrating this information with molecular dynamics simulations and genome-scale models of metabolism provided further insight into the molecular effects of these mutations and how they impacted cellular networks. We conclude that mutations affecting CS activity, and more broadly flux through the TCA cycle and glyoxylate shunt, were instrumental for potentiating the evolution of Cit$^{+}$ , and then for refining the Cit$^{++}$ phenotype. Our results underscore two principles of the evolution of complex systems that operate on long timescales. First, certain mutational paths that are immediately adaptive because they improve fitness in the current niche may be fortuitously co-opted to make future innovative leaps to new traits possible. Second, evolutionary innovations will often have disruptive effects on cellular networks, inverting epistatic relationships and prompting new waves of adaptation that may even reverse the effects of mutations that were necessary for accessing an innovation in the first place.

## Results

### Citrate synthase mutations occur before and after citrate utilization evolves

In a previous study, genome sequencing and phylogenetic analysis of clones isolated from the Ara–3 LTEE population revealed multiple mutations in the gene for citrate synthase (*gltA*) (*Blount et al., 2012*). The earliest evolved allele (*gltA1*) is a single base change that causes an amino acid substitution (A258T) in this enzyme (*Figure 1A*). This mutation was present in every Cit$^{+}$ strain and in earlier Cit$^{-}$ clones from the clade that gave rise to Cit$^{+}$. Therefore, the *gltA1* mutation arose before the Cit$^{+}$ phenotype evolved. Three secondary *gltA* mutations (*gltA2* alleles) were also found, separately, in the genomes of different Cit$^{+}$ clones (*gltA2-4*, *gltA2-6*, or *gltA2-7*). Each of these base substitutions causes an additional single amino acid change in the citrate synthase protein sequence (A124T, V152A, or A162V, respectively).

To better understand the timing and diversity of *gltA* mutations in the LTEE, we performed metagenomic sequencing on Ara–3 population samples spanning a period from 2000 to 45,000 generations (*Figure 1B*). The earliest we detected the *gltA1* mutation was at 25,000 generations. Curiously, this allele dropped to a frequency within the population below the level of detection (~1–5%) in the next sample analyzed at 30,000 generations. As expected, the frequencies of mutations characteristic of diverged clades that never evolved Cit$^{+}$ proportionally increased during this time. The *gltA1* allele emerged again at 33,000 generations and reached nearly 100% frequency in the population by 33,500 generations. This resurgence coincided with the rise of the newly evolved *gltA1*-containing Cit$^{++}$ subpopulation that had also accumulated the *citT* and *dctA\** mutations necessary for robust citrate utilization by this time (*Quandt et al., 2014*). These *gltA1* allele dynamics suggest that the clade that gave rise to Cit$^{+}$ was rare within the population for several thousand generations prior to the evolution and refinement of this metabolic innovation that enabled it to achieve numerical dominance.

During and after the expansion of the Cit$^{++}$ subpopulation, a diverse array of secondary *gltA* alleles arose (*Figure 1*). Two of the three *gltA2* mutations found in the sequenced clones and six other *gltA2* mutations were present in ≥5% of the population at some point. All of these additional

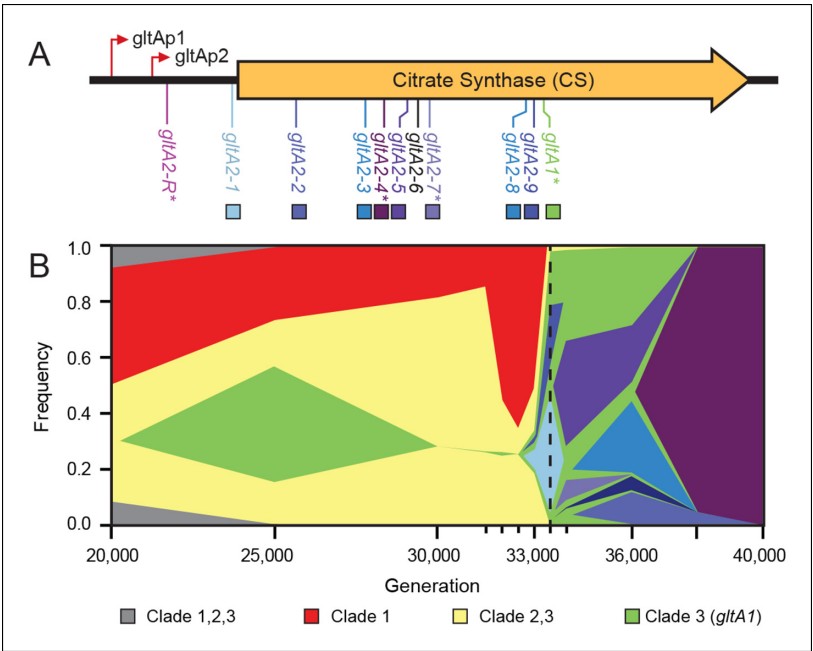

**Figure 1.** Diversity and dynamics of *gltA* mutations in the *E. coli* population that evolved citrate utilization during the Lenski long-term experiment (LTEE). (**A**) Mutations observed in the citrate synthase protein-coding sequence or upstream intergenic region. The *gltA1* mutation evolved first in the LTEE. Many secondary *gltA* mutations later evolved independently in the LTEE (numbered *gltA2* alleles) or in a previous genetic study (*gltA2-R*). All *gltA2* mutations occurred in genetic backgrounds that included *gltA1* and the *citT* and *dctA\** mutations that confer robust citrate utilization (***Quandt et al., 2014***). The effects of starred *gltA* mutations on citrate synthase activity were experimentally characterized in this study. Transcription start sites are labeled gltAp1 and gltAp2 (***Wilde and Guest, 1986***). See ***Figure 1—source data 1*** for the DNA and protein sequence changes caused by each *gltA* mutation. (**B**) Muller plot of evolved allele frequencies over time constructed by using metagenomic DNA sequencing to profile archived samples of the LTEE population. Shaded regions correspond to the frequencies of genetically diverged subpopulations distinguished by different evolved alleles. When a colored sector arises within another region, it is a descendant of that genotype with at least this one new mutation and all of the mutations present in the earlier genotype because these populations are strictly asexual. Clades 1–3 correspond to a phylogenetic tree previously constructed by sequencing the genomes of clonal isolates (***Blount et al., 2012***), which also constrains the order of certain mutations (e.g., *gltA1* < *citT* < *dctA\** < any *gltA2*). Each of the *gltA2* alleles (shaded as in panel A) evolved independently and defines a separate evolved genotype and all of its descendants. Sequencing was performed on population samples from the times indicated with tick marks, and allele dynamics are shown as linearly interpolated between these points. The cell density of the entire LTEE population increased just prior to the dashed line at 33,500 generations (***Blount et al., 2008***) when refined Clade 3 Cit$^{++}$ genotypes containing the *citT* and *dctA\** mutations evolved (***Blount et al., 2012***). ***Figure 1—source data 2*** shows the mutations used to track the different clades.

The following source data is available for Figure 1:

**Source data 1.** Details for all *gltA* alleles in this study.

**Source data 2.** Mutations used to track clade dynamics in the LTEE.

---

*gltA2* alleles are also nonsynonymous mutations, except for one that is a single-base substitution 14 base pairs upstream of the *gltA* start codon. The first *gltA2* mutation (*gltA2-1)* was detected at 33,000 generations. This mutation was apparently completely displaced within ~1000 generations by competition with various other genetically diverged subpopulations, some of which had other *gltA2* alleles. This period of coexistence lasted ~5000 generations with the *gltA2-3* and *gltA2-5* alleles each dominating for a time. By 38,000 generations, the *gltA2-4* mutation had swept to fixation within the population, and no new *gltA* mutations were observed after 40,000 generations.

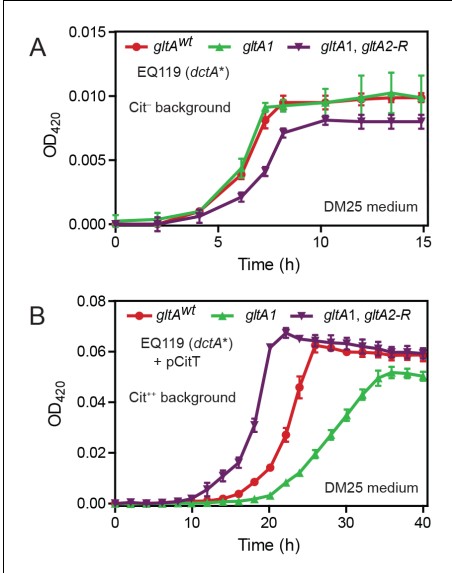

**Figure 2.** Effects of *gltA* alleles on growth depend on the nutrient utilization niche. (**A**) Growth curves of Cit– strain EQ119 containing just the evolved *dctA** mutation and EQ119-derived strains reconstructed with evolved *gltA* alleles. Growth was measured in the DM25 medium used in the LTEE, which contains 0.0025% glucose (w/v) and 0.032% citrate (w/v). These results show that *gltA1* allele has no significant effect on growth when placed into a Cit– genetic background. Further addition of the *gltA2-R* mutation that evolved in a Cit++ genetic background greatly inhibits growth in this genetic background that cannot utilize citrate. Error bars are the S. D. of at least three replicates. (**B**) Growth curves of the same EQ119-derived strains transformed with plasmid pCitT in DM25 medium. pCitT contains the activated *rnk-citT* promoter configuration that evolved in the LTEE; it enables strains like EQ119 with the *dctA** mutation to fully utilize citrate (Cit++ phenotype) (*Quandt et al., 2014*). In the Cit++ genetic context, the *gltA1* allele that evolved in a Cit– genetic background in the LTEE is very deleterious to growth. However, adding the *gltA2* allele to this genetic background suppresses the deleterious effect of the *gltA1* allele and results in further improved growth characteristics compared to the strain containing the *gltAwt* allele. Error bars are the S. D. of at least three replicates.

In a separate experiment, an early Cit+ 33,000-generation LTEE clone with the *gltA1* mutation, but no *gltA2* mutation, was genetically backcrossed with the LTEE ancestor strain using recursive genome-wide recombination and sequencing (REGRES) to determine which evolved alleles were necessary for the Cit+ phenotype (*Quandt et al., 2014*). This procedure involved several rounds of selecting for colony growth on agar containing citrate as the only carbon source. We found a de novo point mutation 182 base pairs upstream of the *gltA* start codon (*gltA2-R* allele) in every one of the sequenced REGRES clones that retained the *gltA1* mutation present in the initial Cit++ clone (*Quandt et al., 2014*). These results suggest that a secondary *gltA2* mutation was also needed for robust citrate utilization under these conditions.

## Citrate synthase mutations were beneficial when they evolved

The prevalence of *gltA* mutations, and in particular of so many secondary *gltA* mutations in Cit++ isolates, suggests that altering citrate synthase activity was beneficial during the LTEE. To directly test the effects of the evolved *gltA* alleles on growth, we created isogenic strains in which we introduced only the *gltA1* mutation or both the *gltA1* and *gltA2-R* mutations into the genome of strain EQ119. This strain (EQ119) was created by reconstructing the *dctA** refinement mutation required for Cit++ in the LTEE ancestor strain REL607. EQ119 is not able to grow on citrate on its own, but it becomes capable of robust growth on citrate when supplied with the genetic module containing the *citT* gene and its captured promoter on a low copy plasmid (pCitT) (*Quandt et al., 2014*).

We compared growth of these strains in DM25, the citrate-containing glucose minimal medium used throughout the LTEE. Without the pCitT plasmid, and therefore without access to citrate, cells containing the *gltA1* mutation grew indistinguishably from EQ119 cells with a fully wild-type citrate synthase sequence (*gltAwt* allele) (*Figure 2*). However, cells with both the *gltA1* and *gltA2-R* mutations had somewhat slower growth rates and reduced final cell densities compared to the strain with only the *gltA1* mutation, indicating that addition of the *gltA2-R* is deleterious when citrate cannot be used as a carbon and energy source.

When made Cit++ via transformation with the pCitT plasmid, EQ119 cells with the *gltA1* mutation grew noticeably worse than those with the *gltAwt* allele, displaying both a longer lag phase and a slower exponential growth rate (*Figure 2*). Addition of the *gltA2-R* mutation in this context relieved the defect caused by *gltA1* and even resulted in improved growth compared to cells with a wild-type citrate synthase sequence. Therefore, the *gltA1* mutation, while seemingly having little effect on glucose growth, was strongly deleterious in Cit++ cells. These results suggest that other

gltA2 alleles may similarly compensate for the gltA1-mediated growth defect and refine a Cit++ cell's ability to rapidly utilize citrate.

Most mutations that accumulate during the LTEE improve fitness (**Barrick et al., 2009**), so it was unexpected that the gltA1 mutation did not appear to appreciably impact growth in the initial growth curve experiment. However, the rate of adaptation in the LTEE slows over time, such that the typical beneficial mutations occurring at later generations only slightly improve competitive fitness (**Wiser et al., 2013**). Therefore, it is possible that growth curves are not sensitive enough to detect a small, but highly relevant effect of the gltA1 mutation on fitness. Alternatively, the gltA1 mutation might only improve fitness in the specific genetic background or ecological context that existed when it evolved during the LTEE.

To discriminate between these possibilities, we performed co-culture competition experiments with strains ZDB478 and ZDB483, two Cit⁻ clones with the gltA1 mutation that were isolated from the LTEE population at 25,000 generations (**Blount et al., 2012**). Specifically, we pitted each of these strains against its respective isogenic derivative with the gltA gene sequence reverted to the ancestral allele (gltA^wt). In DM25, the evolved gltA1 allele had no detectable effect on the fitness of strain ZDB478, but it did improve the competitive fitness of strain ZDB483 by ~3.5% (**Figure 3A**).

E. coli excretes acetate as an overflow metabolite during growth on glucose and then switches to utilizing this acetate after glucose is depleted (**Wolfe, 2005**). Citrate synthase is a component of the glyoxylate pathway, the primary route by which acetate is assimilated in E. coli. Therefore, we reasoned that gltA mutations might specifically improve a component of fitness related to growth on this byproduct. We tested this hypothesis by measuring the effect of gltA1 on the fitness of strains ZDB478 and ZDB483 in DM25 medium supplemented with acetate to simulate an ecological context in which this resource transiently accumulates. We found that the gltA1 mutation increased fitness in both strains under these conditions (**Figure 3A**). Other mutations present in these evolved clones are required for gltA1 to have these beneficial effects, as we found that adding gltA1 on its own to the ancestral REL606 strain led to a significant growth defect in DM25 supplemented with acetate (**Figure 3B**). Altogether, these competition results suggest that the gltA1 mutation was beneficial in the genetic background and nutrient context in which it evolved, specifically because it improved acetate utilization.

## Mutations in glyoxylate and TCA cycle regulators improve acetate utilization

The dependence of the fitness effect of the gltA1 mutation on acetate and an evolved genetic background prompted us to search for other mutations that might have arisen as part of a larger adaptive network for acetate utilization. We identified mutations causing amino acid substitutions in IclR (L201R) and ArcB (Q79L). IclR is a negative regulator of the aceBAK operon (**Cortay et al., 1991**; **Maloy and Nunn, 1982**), which encodes enzymes for the bypass segment of the glyoxylate pathway (**Kornberg, 1966**). Derepression of this operon has been shown to increase acetate utilization (**Maloy and Nunn, 1981**), reduce acetate excretion (**Farmer and Liao, 1997**), and decrease lag time when switching from glucose to acetate in mixed media (**Spencer et al., 2007**). ArcB is the sensor kinase of the ArcAB two-component system, which negatively regulates the expression of genes encoding enzymes in the TCA and glyoxylate cycles, including gltA and the aceBAK operon (**Iuchi and Lin, 1988**; **Waegeman et al., 2011**). Introduction of these mutations into the LTEE ancestor strain REL607 greatly increased its proficiency for acetate assimilation, as judged by growth curves (**Figure 3**). Therefore, it seems likely that both the evolved iclR and arcB alleles are loss-of-function mutations that derepress expression of enzymes needed for acetate assimilation.

## Initial citrate synthase mutation potentiated Cit⁺evolution

To determine whether the initial gltA mutation impacted the evolution of Cit⁺, we compared the effects of the citT mutation in otherwise isogenic pairs of strains with and without the gltA1 allele. Strain ZDB564 was previously isolated from generation 31,500 of the LTEE; it is an early and only weakly Cit⁺ clone containing the citT duplication, but not the dctA* refinement mutation. ZDB564 contains the evolved gltA1 allele shared by all Cit⁺ strains but has no gltA2 mutation. We isolated ZDB706, a spontaneous Cit⁻ revertant of ZDB564 in which its unstable citT duplication had collapsed back to the wild-type sequence. We next reverted the gltA1 mutation to the wild-type sequence

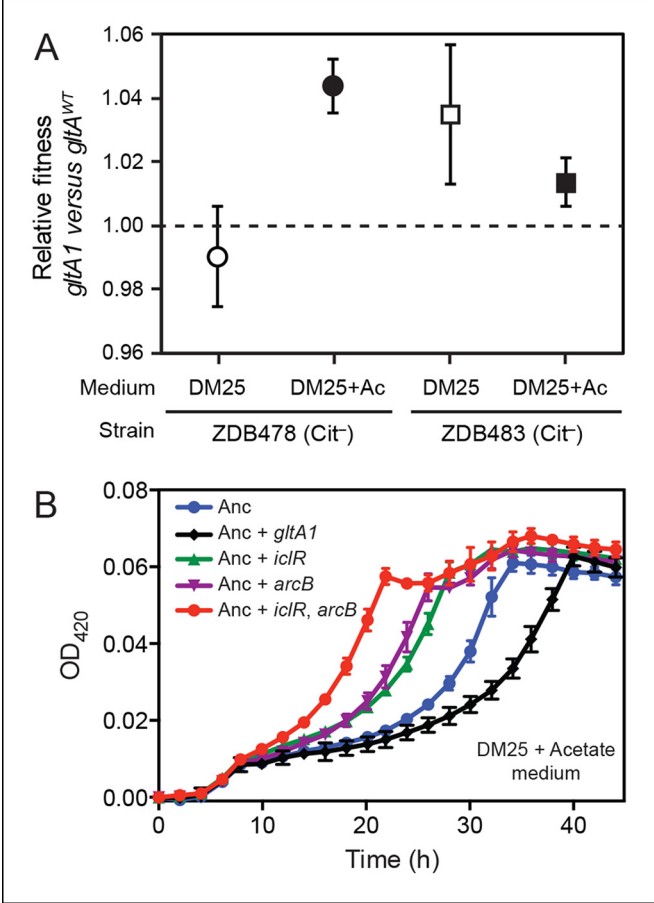

**Figure 3.** Evolved *gltA1*, *iclR*, and *arcB* alleles improve acetate utilization. (**A**) Fitness measurements for two Cit–strains (ZDB478 and ZDB483) isolated from the LTEE population at 25,000 generations with the *gltA1* mutation relative to isogenic strains with this mutation reverted to the wild-type sequence (*gltA^wt^*). The first set of co-culture competition assays was performed in DM25 under normal LTEE conditions. The second set was performed in DM25 supplemented with 0.0025% acetate (Ac) (w/v) to test whether *gltA1* affected the component of fitness related to utilization of this overflow product of metabolism that transiently accumulates during *E. coli* growth on glucose. The presence of the *gltA1* mutation is beneficial to fitness in DM25 in one of these two strains, each of which contains other mutations that evolved before and after *gltA1* during the LTEE. With added acetate the evolved *gltA1* allele is beneficial in both strains, suggesting that this mutation in citrate synthase is important for improving acetate utilization. Error bars are 95% confidence limits from six replicate assays. (**B**) Growth curves of the ancestral REL607 strain (Anc) and derivative strains constructed to contain evolved *gltA1*, *iclR*, and *arcB* alleles. Strains were grown in DM25 media supplemented with 0.05% (w/v) acetate. The *iclR* and *arcB* genes encode transcriptional regulators of the glyoxylate and TCA cycles. These metabolic pathways are required for acetate assimilation in *E. coli*. The evolved *gltA1* mutation is deleterious on its own in the ancestral strain background under these conditions, showing that its beneficial effect on acetate utilization in (**A**) is dependent on other mutations present in ZDB478 and ZDB483. In contrast, mutations in *iclR* and *arcB* improve growth on acetate individually and in combination with one another in the ancestral genetic background. Thus, all three of these mutations likely evolved in the LTEE population because they specifically improved utilization of the acetate byproduct of glucose metabolism. Error bars are the S. D. of at least three replicates.

(*gltA^wt^*) in both strain backgrounds. We were then able to compare differences in growth and fitness between ZDB564 and ZDB706 variants with and without the evolved *gltA1* allele, and thereby test whether this citrate synthase mutation potentiated the evolution Cit⁺ by altering the effects of the *citT* actualizing mutation.

The presence of the *citT* duplication had little effect on the growth of the Cit⁺ /Cit⁻ strain pair containing the *gltA1* allele (*Figure 4*). If anything, the *citT* mutation appeared to be slightly beneficial in terms of growth rate and final cell density. In stark contrast, the *citT* mutation greatly

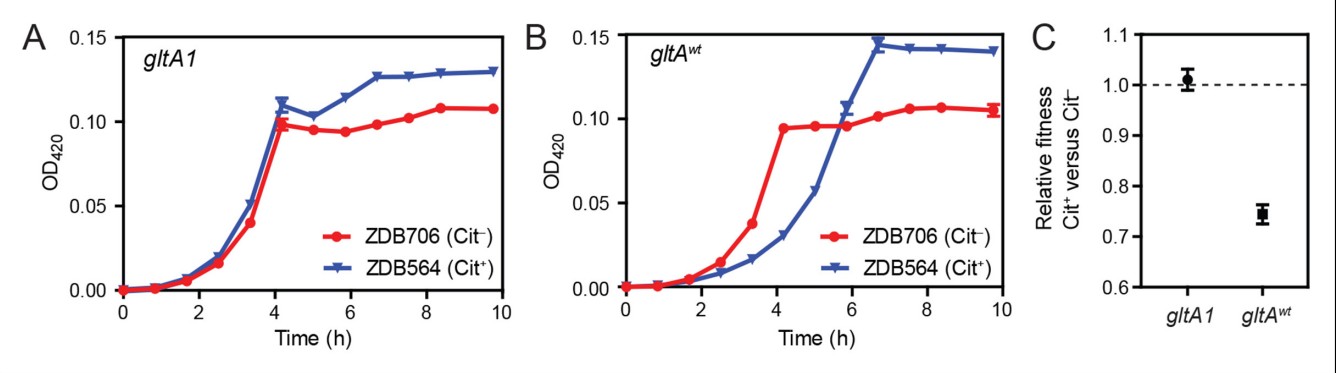

**Figure 4.** Evolution of citrate utilization was potentiated by the *gltA1* mutation. (**A**) Growth curves for an early Cit+ clone (ZDB564) from the LTEE which contains the *gltA1* mutation and an isogenic Cit– revertant of this strain (ZDB706) without the *citT* amplification, grown in DM250 medium. The presence of the *citT* mutation, which is sufficient for the rudimentary Cit+ trait on its own, slightly improves the growth dynamics and final cell density in this genetic background that includes *gltA1* and other evolved alleles. Error bars are the S. D. of at least three replicates. (**B**) Growth curves of isogenic derivatives of strains ZDB564 and ZDB706 in which the *gltA1* mutation has been reverted to the wild-type sequence, performed in DM250 medium. Addition of the *citT* amplification in this *gltA*wt genetic background now causes a large lag in growth dynamics although the final cell density achieved is still higher in the Cit+ strain. Error bars are the S. D. of at least three replicates. (**C**) Relative fitness values as determined by competition assays between Cit+ (ZDB564) and Cit– (ZDB706) isogenic strain pairs which contain either the *gltA1* or *gltA*wt allele. Competitions were performed in DM25 medium, which was used throughout the LTEE. In accordance with the growth curves, these competitions show that the *citT* mutation would be highly deleterious to fitness if it evolved in a genetic background without the *gltA1* mutation. In contrast, the *citT* mutation is neutral or possibly slightly beneficial to fitness when the *gltA1* mutation is present. Thus, the *gltA1* mutation potentiated the eventual evolution of robust citrate utilization (Cit++) by preventing the *citT* mutation from having deleterious effects when it first appeared. Error bars are 95% confidence limits from six replicate assays.

prolonged the lag phase before exponential growth in the pair of strains with the *gltA*wt allele (*Figure 4*). The long lag in growth would have significantly disadvantaged this strain within the LTEE population, even though it can reach a slightly higher final cell density when grown in isolation. Co-culture competition experiments between each pair of Cit+/Cit– strains confirmed that whereas the *citT* mutation was approximately neutral with respect to fitness in the *gltA1* strain background, it was highly deleterious in the *gltA*wt background (*Figure 4*). We conclude that the *gltA1* mutation was quantitatively necessary for potentiating the evolution of citrate utilization in the LTEE population. Epistatic interactions between *gltA1* and the *citT* mutation in this evolved genetic background prevented a massive fitness defect that would have almost certainly led to the rapid extinction of any newly evolved Cit+ cells before this rudimentary trait could be refined to the advantageous Cit++ phenotype by further mutations.

## One secondary citrate synthase mutation reduces mRNA expression levels

We next sought to characterize the effects of the *gltA* mutations on cellular physiology in order to understand their impacts on cellular fitness and interactions with other evolved mutations. We first examined whether the *gltA1* mutation or the *gltA2-R* mutation, which is located in the upstream intergenic region, altered citrate synthase mRNA expression (*Figure 5*). We found that *gltA1* had no effect on transcript levels in this near-ancestral genetic background. In contrast, addition of the *gltA2-R* mutation reduced *gltA* mRNA levels by approximately tenfold, whether or not the *gltA1* mutation was also present. We conclude that the *gltA2-R* mutation targets the *gltA* promoter, and that the fitness effects of this mutation result from reduced gene expression.

## Most secondary citrate synthase mutations reduce enzyme activity

To understand the effects of the other *gltA* mutations that evolved in the Ara–3 population, we examined the enzymatic activity of citrate synthase (GltA). We characterized His6-tagged GltA protein with just the *gltA1* mutation (A258T), as well as variants also containing secondary *gltA2* amino acid substitutions. We found that the A258T substitution did not significantly alter the kinetic parameters of the enzyme (*Table 1*). In contrast, further addition of the *gltA2-7* (A162V) or *gltA2-4* (A124T)

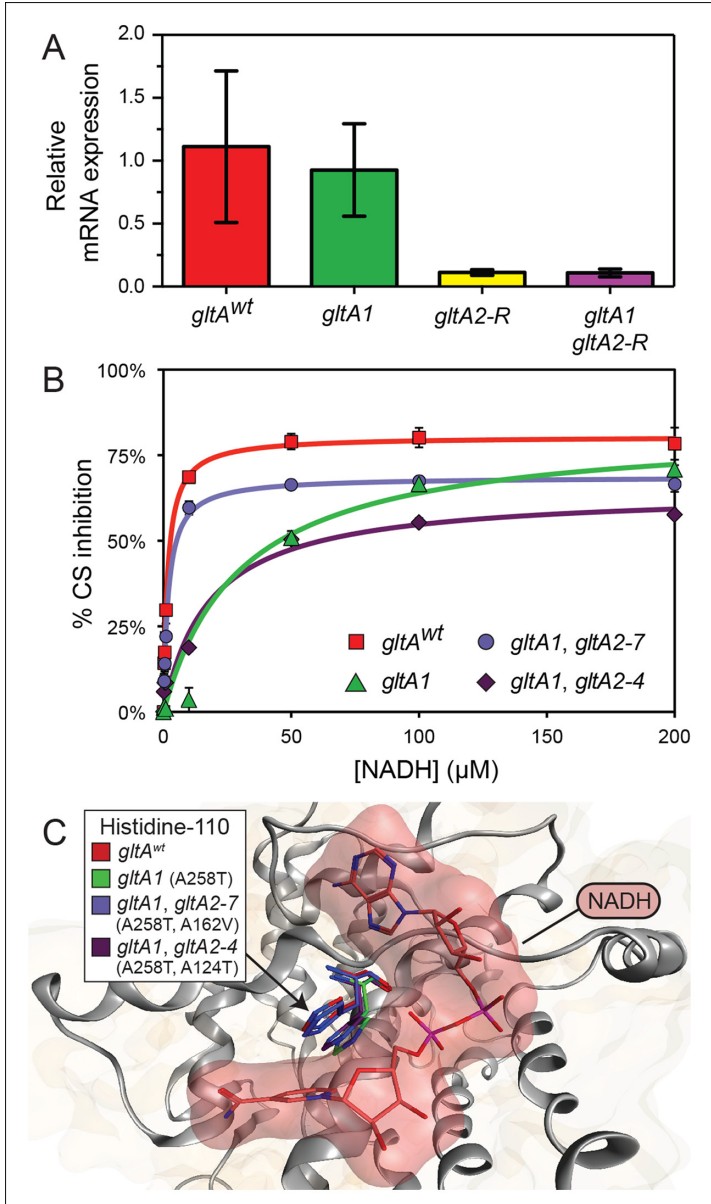

**Figure 5.** *gltA* mutations alter gene expression and allosteric regulation by NADH. (**A**) *gltA* mRNA expression levels as determined by qRT-PCR of EQ119-derived cells containing the specified ancestral or evolved alleles. Expression levels are shown relative to that of strain EQ119 which contains the *gltA*^wt^ gene sequence. Error bars are 95% confidence intervals of biological triplicate samples. (**B**) NADH-mediated inhibition citrate synthase activity for the wild-type enzyme and evolved variants with combinations of *gltA* mutations. Fits to the hyperbolic model used to extract the binding and inhibition parameters in *Table 1* are shown. Error bars are the S. E. M. of three replicates. (**C**) Molecular modeling predicts that the observed changes in allosteric regulation in evolved CS sequences are primarily caused by how mutations affect the orientation of histidine-110 in the NADH binding pocket. The *gltA1* mutation is predicted to redirect this side chain into the binding pocket creating a steric clash with NADH (red surface). The two characterized variants of citrate synthase with an additional *gltA2* mutation are predicted to reorient histidine-110 back toward the wild-type conformation. The degree of this predicted structural change correlates with the relative levels of NADH inhibition experimentally measured for these CS variants (*Table 1*).

**Table 1.** Kinetic and regulatory properties of evolved citrate synthase variants.

| Evolved alleles | Substitutions | $k_{cat}$(s$^{-1}$) | $K_m$ OAA (μM) | $K_m$ AcCoA (μM) | $K_d$ NADH (μM) | $K_i$ NADH (μM) | Maximum% inhibition | Docking energy (kcal/mol) |
|---|---|---|---|---|---|---|---|---|
| gltA$^{wt}$ | none | 51.1 ± 2.5 | 84.2 ± 10.4 | 138.6 ± 17.6 | 1.2 ± 0.1 | 1.7 ± 0.1 | 80.5 ± 1.0 | –80.3 |
| gltA1 | A258T | 49.1 ± 5.3 | 90.3 ± 22.5 | 148.0 ± 41.1 | N. D. | 30.2 ± 4.6* | 83.2 ± 3.1 | –28.8 |
| gltA1 gltA2-7 | A258T, A162V | 55.0 ± 3.0 | 143.4 ± 11.1* | 374.1 ± 41.2* | 1.5 ± 0.4 | 1.9 ± 0.1 | 68.7 ± 0.7* | –63.8 |
| gltA1 gltA2-4 | A258T, A124T | 33.9 ± 2.7* | 88.1 ± 12.8 | 295.1 ± 52.6* | 9.5 ± 1.7* | 17.1 ± 6.2* | 64.6 ± 5.1* | –47.8 |

Data are represented as fit mean ± S. E. Significant differences from the wild-type enzyme are marked with an asterisk (two-tailed t-test, p-value < 0.05). N. D. indicates no detectable binding.

mutations reduced citrate synthase activity. Specifically, the A162V substitution increased the $K_m$ values for oxaloacetate (OAA) (143.4 ± 11.1 μM vs 84.2 ± 10.4 μM) and acetyl-CoA (AcCoA) (374.1 ± 41.2 μM vs 138.6 ± 17.6 μM), the two substrates of citrate synthase. The A124T amino acid substitution resulted in an increased $K_m$ for acetyl-CoA (295.1 ± 52.6 μM vs 138.6 ± 17.6 μM) and a reduction in the $k_{cat}$ value (33.9 ± 2.7 s$^{-1}$ vs 51.1 ± 2.5 s$^{-1}$). We conclude that *gltA2* mutations likely impact cellular metabolism in a similar manner, as they all reduce citrate synthase activity. This reduction is achieved either by affecting the Michaelis-Menten parameters of the enzyme (*gltA2-7* and *gltA2-4*) or by lowering the steady-state level of the *gltA* transcript, and thus the expression of the protein (*gltA2-R*).

## The initial *gltA* mutation (*gltA1*) diminishes allosteric inhibition by NADH

Given that the *gltA1* mutation (A258T) did not significantly alter mRNA levels or Michaelis-Menten enzyme parameters, we hypothesized that it might affect allosteric regulation of the enzyme. *E. coli* GltA is a type II citrate synthase that is inhibited by NADH, the primary product of the TCA cycle (*Weitzman, 1966a*, *1966b*; *Weitzman and Jones, 1968*). NADH binding to purified GltA proteins was measured via changes in NADH fluorescence that occur upon associating with the enzyme (*Duckworth and Tong, 1976*), and allosteric inhibition was measured by analyzing kinetic parameters determined in the presence of different NADH concentrations. Wild-type GltA was found to exhibit a $K_d$ and a $K_i$ for NADH of ~1 μM (*Table 1* and *Figure 5*), in excellent agreement with earlier reports (*Anderson and Duckworth, 1988*; *Pereira et al., 1994*; *Stokell et al., 2003*). However, the *gltA1* (A258T) mutation greatly diminished NADH binding, resulting in an inability to saturate the enzyme within the range of detection in the binding assay, which extends up to ~10 μM NADH (*Dickinson, 1970*). Likewise, the *gltA1* mutant displayed a $K_i$ for NADH that was higher by a factor of ~30 relative to wild-type enzyme. Each *gltA2* mutation restored NADH binding and allosteric inhibition, either partially or fully. Notably, GltA containing both the *gltA1* and the *gltA2-7* mutations (A258T, A162V) displayed near wild-type $K_d$ and $K_i$ values for NADH (*Table 1*).

To gain further insight into how these mutations affected allosteric inhibition by NADH, we performed molecular dynamics simulations. Mutations were introduced into the structure of *E. coli* citrate synthase (*Maurus et al., 2003*). After energy minimization, we observed subtle differences in the predicted conformation of each enzyme variant, most notably around the NADH binding site. The most pronounced changes were in the orientation of histidine-110 (*Figure 5*), even though this amino acid is distant from every *gltA* mutation. In the wild-type GltA structure, histidine-110 adopts an upward conformation that allows for accommodation of NADH in the binding pocket. The A258T *gltA1* mutation is predicted to reorient the histidine-110 side chain toward the binding pocket, presumably creating an unfavorable steric barrier to NADH binding. Addition of secondary *gltA2* mutations (A162V or A124T) resulted in simulated structures with histidine-110 oriented between these two extremes. Computational docking of NADH to these mutant CS structures predicted relative binding energies that were correlated with the experimentally determined NADH binding affinities (*Table 1*).

## Metabolic modeling predicts the evolutionary reversal in citrate synthase activity

Having determined the molecular consequences of *gltA* mutations, we next sought to use flux balance analysis (FBA) (*Orth et al., 2010*) to evaluate how changes in citrate synthase activity would affect cellular growth rates. When the native regulatory pathways of bacteria cannot adjust enzyme activity to achieve optimal reaction fluxes, mutations—like those we observed in *gltA*—that break these constraints may be necessary to maximize growth rates (*Ibarra et al., 2002*; *Lewis et al., 2010*; *Teusink et al., 2009*). We used the metabolic model for the LTEE ancestor strain (*Monk et al., 2013*) to predict reaction fluxes that are optimal for growth on glucose, citrate, or acetate (*Figure 6*). In addition, we also examined how constraining CS flux would impact growth on each of these substrates (*Figure 7*).

For growth on glucose, FBA predicts that low citrate synthase flux will limit the rate of biomass accumulation (*Figure 7*). This observation agrees with the known importance of citrate synthase in the biosynthesis of glutamate in glucose minimal media (*Davis and Gilvarg, 1956*). However, when citrate is the sole carbon source, FBA predicts optimal growth when there is no flux through CS (*Figure 7*). In fact, any CS activity is detrimental because sufficient flux to synthesize glutamate (and other compounds) from TCA cycle intermediates is already available. Under these conditions,

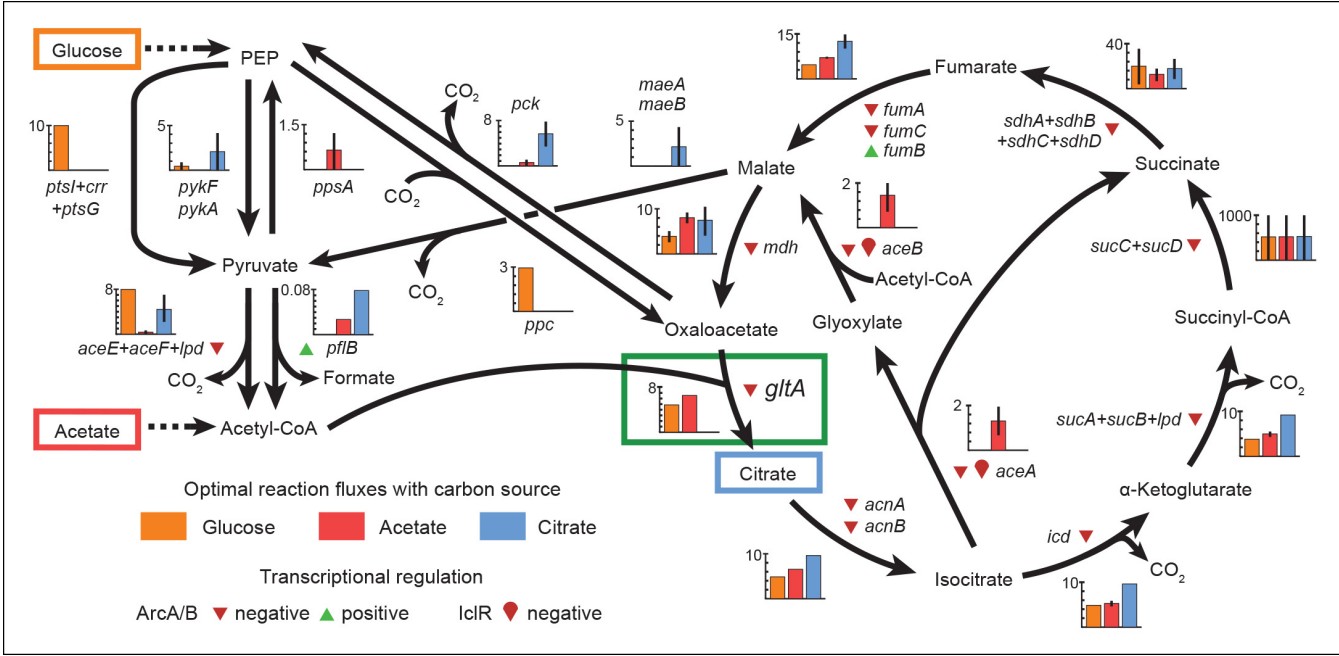

**Figure 6.** Optimal reaction fluxes on carbon sources present in the *E. coli* LTEE. Flux balance analysis (FBA) was used to predict the reaction fluxes in the *E. coli* B REL606 ancestral strain of the LTEE that optimize the rate of biomass accumulation when utilizing a single carbon source: either glucose, acetate, or citrate. Glucose is the primary carbon source for cells grown under LTEE conditions, while acetate, a metabolic overflow product excreted during *E. coli* growth on glucose, can also be utilized by the ancestral strain (*Yoon et al., 2012*). Citrate can only be utilized under the aerobic conditions of the LTEE after the Cit+ innovation (*Blount et al., 2008*, *2012*). Flux values derived from FBA for key reactions in central metabolism are shown for modeling growth on each carbon source: glucose (orange bars), acetate (red bars), and citrate (blue bars). Error bars show the full range of possible fluxes for each reaction that are consistent with globally optimal FBA solutions, as predicted by flux variability analysis, and the colored bars show an intermediate value in each range. Gene names for all enzymes that contribute to each reaction flux are displayed. Genes whose expression is controlled by IclR and ArcAB are marked with symbols indicating the direction of transcriptional regulation, and the key citrate synthase (CS) reaction catalyzed by GltA is boxed in green. The results of the FBA modeling agree with our experimental observations that the combined effects of derepressing the IclR and ArcAB regulons via the *iclR* and *arcB* mutations and alleviating NADH-mediated allosteric inhibition of CS via the *gltA1* mutation are beneficial for growth on acetate because they increase flux values for these reactions toward levels that are optimal for this substrate. This metabolic program is thought to potentiate the evolution of Cit+ by increasing the production of $C_4$-dicarboxylates (succinate, fumarate, and malate) which can be exported in exchange for citrate uptake by the CitT antiporter. Under Cit++ conditions in which citrate is the primary carbon source FBA predicts that drastically reducing flux through the CS reaction is required to achieve an optimal growth rate which also agrees with our finding that beneficial *gltA2* mutations that decrease CS activity evolved at this point in the LTEE.

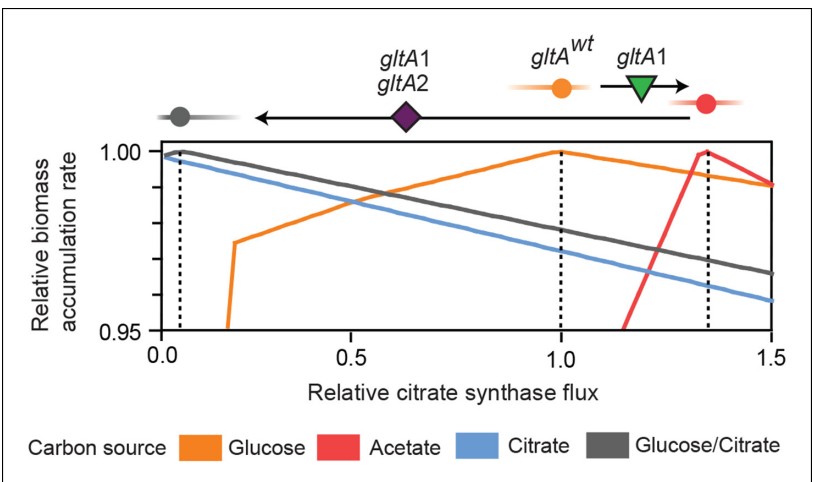

**Figure 7.** Effects of constrained citrate synthase flux on utilizing different carbon sources available during the LTEE. Rates of biomass accumulation on different substrates were calculated by using FBA to optimize metabolic fluxes subject to a defined constraint on citrate synthase flux. Glucose/citrate represents a growth condition with the molar ratio of those two nutrients that is present in the DM25 medium used in the LTEE. Curves for other carbon sources are colored as in *Figure 7*. Relative biomass accumulation rates were normalized within each curve to the maximum value achieved on the specified substrate or substrate mixture. Relative citrate synthase flux was normalized to the value that resulted in maximum biomass accumulation rate for glucose growth. Optimal relative flux values for each carbon source are indicated with dashed lines. Arrows above each graph indicate the changes in CS activity expected from each successive mutation in *gltA* that evolved in the LTEE. This analysis shows how mutations in *gltA* appear to have arisen because they enable flux through CS to achieve values that are more optimal for growth, first on the acetate byproduct of glucose metabolism (*gltA1*) and then on the glucose/citrate mixture that can be metabolized once robust aerobic citrate utilization has evolved (*gltA1 + gltA2*).

oxaloacetate (OAA) is diverted into gluconeogenesis for the production of essential glycolytic intermediates and sugars (*Sauer and Eikmanns, 2005*). This shift in metabolism leads to FBA predicting an increase in flux through the phosphoenolpyruvate (PEP) carboxykinase (*pck*) and malic enzyme (*maeA/maeB*) reactions (*Figure 6*). Eliminating CS flux also preserves its other CS substrate, acetyl-CoA, so that it can be used for other biosynthetic processes, rather than to produce unneeded TCA intermediates. Cit++ cells must balance using both glucose and citrate, so the optimal level of CS activity in these cells is predicted to be low but not zero (*Figure 7*), consistent with the effects of the *gltA2* mutations we studied.

Since acetate is initially excreted during glucose growth and then subsequently utilized after glucose is depleted, it can represent a distinct resource niche in the LTEE, as described above. FBA indicates that optimal growth on acetate occurs when the flux through citrate synthase is approximately 30% higher than the level that is optimal for growth on glucose (*Figure 7*). This increase is expected because CS is needed for the glyoxylate cycle, the major pathway for acetate assimilation in *E. coli* (*Kornberg, 1966*). Thus, FBA predicts that the *gltA1* mutation that increased CS flux improved the component of competitive fitness in the LTEE related to acetate utilization. Flux through the bypass portion of the glyoxylate pathway, comprised of the isocitrate lyase (*aceA*) and malate synthase (*aceB*) reactions, is needed for optimal growth on acetate but not for growth on glucose (*Figure 6*). This FBA prediction supports our hypothesis that the LTEE mutations in *iclR* and *arcB*, which we showed improve growth on acetate-enriched media (*Figure 3*), do so because they derepress transcription of *aceA* and *aceB*.

The FBA predictions motivated us to look for other mutations in central metabolism that may have contributed to refining the Cit++ phenotype. Loss of isocitrate lyase (*aceA*) activity is predicted to be beneficial for growth on citrate (*Figure 6*) because it eliminates unnecessary input of acetyl-CoA into the TCA cycle via the glyoxylate shunt. We found that a nonsense mutation in *aceA* was present in ~15% of the LTEE population in the 33,500-generation sample and ~97% in the next 34,000-generation sample. Thus, nearly all Cit++ cells had either the *aceA*, *gltA2-1*, or *gltA2-9* allele

at 33,500 generations (*Figure 1*). By 34,000 generations the *aceA* mutation had nearly swept to fixation within the Cit$^{++}$ clade and all subsequent *gltA2* alleles occurred in that genetic background. An *aceA* mutation may be especially necessary to adjust isocitrate lyase flux because, as discussed earlier, these strains already contain mutations in the *iclR* and *arcB* transcriptional regulators that appear to compromise the regulatory mechanisms that would normally repress *aceA* gene expression.

## Discussion

We have established that mutations in the *gltA* gene encoding citrate synthase (CS) were critical for both potentiating the evolution of aerobic citrate utilization in the Lenski LTEE and for the subsequent refinement of this new metabolic capability. A key insight from our studies was the importance of fitness components in the LTEE related to utilizing both glucose, the sole carbon source added to the growth medium, and acetate, a transient overflow byproduct of *E. coli* growth on glucose (*Figure 8*). We found evidence that the clade in which Cit$^+$ would eventually evolve had previously evolved a suite of mutations that improved the utilization of acetate. We assert that this particular mutational trajectory in the glucose-acetate fitness landscape reshaped metabolic fluxes in a way that facilitated the transition to aerobic citrate utilization.

Acetate accumulation has previously been shown to reliably lead to the appearance and co-existence of glucose and acetate specialists in shorter *E. coli* evolution experiments (*Herron and Doebeli, 2013*; *Spencer et al., 2007*; *Treves et al., 1998*). Rather than give rise to this type of stable ecological diversification, the much lower concentration of glucose (and therefore acetate) in the LTEE appears to have largely favored the success of generalists that incorporate mutations that improve growth on both substrates. For example, mutations in the transcriptional regulators, *iclR* and *arcB,* arose in the population that evolved citrate utilization before 25,000 generations (*Blount et al., 2012*). These mutations are expected to derepress the transcription of the mRNAs encoding enzymes in the TCA and glyoxylate cycles, which are necessary for assimilating acetate via acetyl-CoA. Similar *iclR* and *arcB/arcA* mutations are found in acetate specialists in other evolution experiments (*Herron and Doebeli, 2013*; *Spencer et al., 2007*).

Interestingly, changes affecting acetate metabolism in the LTEE were not unique to the population that evolved Cit$^+$. Strains isolated at 50,000 generations from all LTEE populations excreted 50% more acetate, on average, than the ancestral strain (*Harcombe et al., 2013*). Mutations in both *iclR* and *arcB* are also present by 15,000 generations in a population that has not evolved Cit$^+$ (*Barrick et al., 2009*), and *arcB/arcA* mutations have been found in 11/12 of the LTEE populations (*Plucain et al., 2014*). As expected from the widespread appearance of mutations in *iclR* and *arcB/arcA*, there was universal improvement in acetate growth for 20,000-generation isolates from all LTEE populations (*Leiby and Marx, 2014*).

In contrast, mutations in *gltA* are rare in the LTEE. Only one mutation in citrate synthase was found among 16 clones isolated at generations 20,000 to 40,000 from 7 other LTEE populations (*Wielgoss et al., 2011*). Flux through CS is highly regulated in wild-type *E. coli*, both at the level of transcription (*Gosset et al., 2004*; *Iuchi and Lin, 1988*; *Park et al., 1994*) and via allosteric feedback inhibition by NADH (*Weitzman, 1966a*, *1966b*), as is typical of gram-negative bacteria (*Maurus et al., 2003*; *Weitzman and Jones, 1968*). The initial *gltA1* mutation in the population that evolved Cit$^+$ disrupted allosteric repression of CS by NADH (*Figure 9*). Increasing CS activity in this way is predicted to improve *E. coli* growth on acetate. We found that the *gltA1* mutation did indeed improve competitive fitness when it was added to a clone isolated from the LTEE population around the time when it appeared (*Figure 8B*), particularly in growth medium supplemented with acetate. As the lineage with *gltA1* seems to have come close to extinction in the Ara–3 population before it evolved efficient citrate utilization, it is possible that specializing towards better acetate utilization gave this lineage a frequency-dependent fitness advantage when rare against other competitors in this population that helped preserve it until Cit$^{++}$ evolved.

We hypothesize that the *gltA1* mutation critically affected the fitness consequence of the pivotal evolutionary step toward innovation: the *citT* mutation that enabled the first citrate import into cells and the weak Cit$^+$ phenotype. An allosterically deregulated citrate synthase enzyme, which continuously inputs increased carbon flux into the TCA cycle during growth on glucose/acetate, coupled with overall transcriptional derepression of the TCA and glyoxylate cycles, could replenish the intracellular supply of succinate and/or other $C_4$-dicarboxylates that are excreted in exchange for citrate

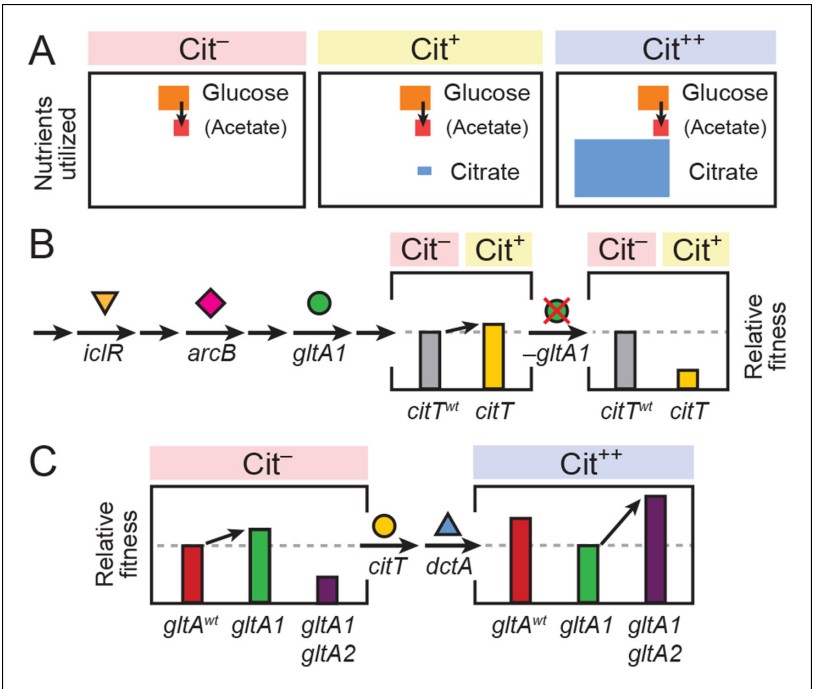

**Figure 8.** Model for how successive mutations in *gltA* are important for first potentiating the evolution of citrate utilization and then for refining this new trait. (**A**) Different carbon sources can be utilized by *E. coli* during each stage in the evolution of aerobic citrate utilization in the Lenski LTEE. Initially only glucose and acetate, a byproduct of glucose metabolism, are accessible to Cit– cells. The rudimentary Cit+ trait enabled very limited utilization of some of the abundant citrate present in the growth medium. Once this ability was refined to the Cit++ phenotype by further mutations, all of the citrate present was fully exploited. The areas of each box roughly reflect the relative amounts of each of the three carbon sources. (**B**) The initial *gltA1* mutation potentiated the evolution of Cit+. This mutation and additional mutations in *iclR* and *arcB* all improve acetate utilization in Cit– cells by increasing flux through the TCA and glyxolyate cycles. The order of accumulation of these mutations, interspersed with many other mutations during the LTEE, is represented by arrows on the left side of the figure. These mutations may potentiate the evolution of the weak Cit+ phenotype because they create a potentiated version of central metabolism that is capable of replenishing succinate and/or other C4-dicarboxylates that must be exported in exchange for citrate import by the CitT antiporter. In accordance with this model, the addition of the CitT-activating mutation (*citT*) to an evolved genetic background that crucially contains the *gltA1* mutation results in a rudimentary Cit+ phenotype that is neutral or slightly beneficial to fitness and thus represents a path accessible to evolution, denoted by an arrow to mark the transition as shown in left bar graph. In contrast, adding the *citT* mutation to the same genetic background but without the *gltA1* mutation is highly deleterious to fitness, rendering this path inaccessible to evolution within the LTEE population, as shown in right bar graph. (**C**) Fitness effects of *gltA* alleles show sign epistasis with nutrient utilization niche. The *gltA1* mutation that increased citrate synthase activity was beneficial to fitness when it evolved prior to the Cit+ innovation when only glucose and acetate could be utilized, as shown in the bar graph on the left. While necessary for the evolution of the rudimentary Cit+ phenotype by *citT* activation, the effect of *gltA1* mutation is suboptimal for fitness once the *dctA*\* mutation evolves and the citrate utilization niche can be fully exploited. Under this condition, the *gltA1* mutation was deleterious to fitness and *gltA2* mutations evolved that reversed the effect of the *gltA1* mutation and even further reduced citrate synthase activity below the ancestral level to improve fitness in this niche, as shown in bar graph on the right. Note that relative fitness is on a different scale in each of the fitness diagrams, as all Cit++ strains are considerably more fit than any Cit– strain due to the abundant citrate in the growth medium.

by the CitT antiporter. The unbalanced loss of these important biosynthetic precursors might explain the detrimental fitness effect of becoming Cit+ via the *citT* mutation in cells lacking *gltA1* (*Figure 8C*). Thus, the *gltA1* mutation was necessary for potentiating the evolution of Cit+ because it converted the *citT* duplication from a prohibitive step downward into a valley in the fitness landscape into a step in an upward mutational route.

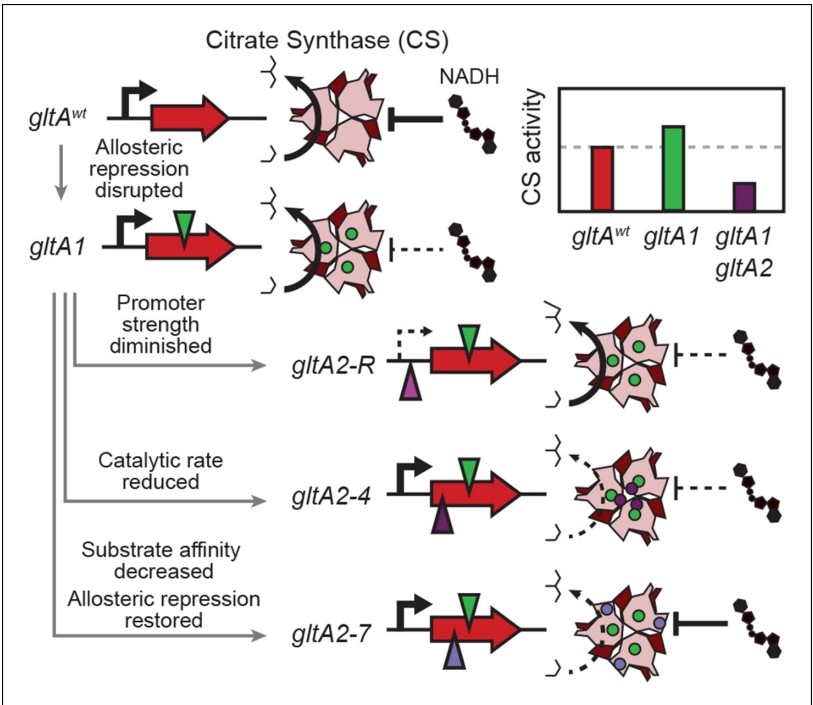

**Figure 9.** Summary of the molecular effects of evolved *gltA* mutations. The molecular effects of the initial *gltA1* mutation and of the various *gltA2* mutations were characterized and the mechanisms by which they produce changes in cellular citrate synthase activity are depicted in the series of gene and protein structure diagrams. Approximate locations of mutations in the *gltA* gene are shown as shaded triangles in the series of gene diagrams and likewise the locations of the resulting amino acid changes are shown as similarly shaded circles on the CS structure diagrams. Reduced transcription, enzyme activity, and allosteric inhibition as compared to wild-type are indicated with dashed lines. The inset shows the approximate relative levels of citrate synthase activity representative of each *gltA* allelic state.

After strong citrate utilization (Cit$^{++}$ phenotype) evolved in the LTEE due to the activation of the DctA transporter by the *dctA** promoter mutation, multiple secondary *gltA2* alleles reached high frequencies in this population in separate lineages vying for dominance. These *gltA2* mutations were beneficial for growth on citrate as the primary carbon source (*Figure 8B*), and they share a common overall effect: all are expected to decrease citrate synthase activity. When growing on citrate as a carbon source, the CS reaction is expected to be detrimental in that it consumes acetyl-CoA and diverts OAA that is needed for gluconeogenesis back into the TCA cycle, a futile-cycle under these conditions. While allosteric regulation of wild-type CS may have been able to adjust flux through this reaction to the very low levels that are optimal under these conditions, this was apparently not possible with the *gltA1* mutation already present. Therefore, *gltA2* mutations emerged that reverse the change in enzyme activity caused by *gltA1*, either by decreasing mRNA expression levels, by reducing the catalytic proficiency of this enzyme, and/or by restoring allosteric inhibition by NADH (*Figure 9*).

To a first approximation, this LTEE population can be thought of as having evolved through three metabolic epochs: first, glucose utilization was optimized, leading to greater acetate accumulation; second, acetate utilization was optimized in conjunction with further improvements in glucose growth; third, citrate utilization was discovered and optimized. As a whole, the LTEE populations have explored numerous variations on the complex metabolic and regulatory networks of *E. coli* as they have adapted. This diversity has allowed some lineages of cells to explore new nutrient niches, in particular citrate utilization. While rudimentary citrate utilization via activation of the *citT* gene could presumably have arisen at any time and in any of the LTEE populations, we have shown that it would have been at a selective disadvantage if it appeared in a non-potentiated genetic background. Moreover, if the structure of the metabolic and regulatory networks and their component

genes yielded relatively few genetic pathways with which to improve growth on glucose, it is unlikely that the multi-step mutational pathway to Cit$^+$ that first required mutations that improve acetate utilization would ever have been realized. Therefore, complexity in both the resource environment and in the genetic architecture of the cell conspired to make this metabolic innovation possible.

More broadly, our results demonstrate that evolutionary innovations may rely not only on the acquisition of novel genes or the co-option of molecular machinery for entirely new purposes, but also on the inherent malleability of core cellular processes. The components of an organism's metabolic, regulatory, and developmental networks have evolved to interact in complex ways that are attuned to its current niche. Yet, these networks are also poised such that they can be dynamically reorganized toward new purposes by only a few mutations in key enzymes and regulatory proteins. As we observed for changes in citrate synthase activity at different stages in the emergence of citrate utilization during the Lenski LTEE, it may often be case that evolution must fine-tune essential links in these networks as it traverses epistatic turns and switchbacks on the tenuous mutational paths that lead to the successful colonization of new niches.

# Materials and methods

## Experimental procedures

### Strains and plasmids

All *E. coli* strains and plasmids used in this study are listed in **Supplementary file 1**. REL607 is an Ara$^+$ derivative of the Ara–3 LTEE ancestor strain REL606 (*Lenski et al., 1991*). Strain EQ119 is an isogenic derivative of REL607 constructed by adding the *dctA\** mutation to its genome (*Quandt et al., 2014*). Strains ZDB483, CZB154, ZDB83, and ZDB107 were isolated from Ara–3 LTEE population samples frozen at generations 25,000, 33,000, 34,000, and 38,000, respectively. Their genomes were sequenced in a previous study (*Blount et al., 2012*). ZDB478 and ZDB564 were isolated at 25,000 and 31,500 generations, respectively. ZDB706 is a spontaneous Cit$^-$ revertant of ZDB564 isolated by propagating this strain in citrate-free media, screening for strains unable to grow on minimal citrate agar, confirming a negative reaction on Christensen's citrate agar, and verifying by PCR that the *citT* amplification had collapsed to its original configuration.

### Growth media

Davis minimal (DM) broth, tetrazolium arabinose (TA) agar, and minimal arabinose (MA) agar have been described elsewhere (*Lenski et al., 1991*). DM25 and DM250 additionally contain 0.0025% and 0.025% glucose (w/v), respectively. Acetate was supplemented at 0.0025% or 0.05% (w/v) where indicated. Lysogeny broth (LB) was of the Lennox formulation (5 g/L NaCl). When appropriate, LB and DM media were supplemented with the antibiotics kanamycin (30 µg/mL) and chloramphenicol (34 µg/mL).

### Metagenomic sequencing

Whole-population samples from the LTEE frozen as glycerol stocks were re-cultured overnight in DM medium with 0.1% glucose. Viable cell counts in the inocula were roughly equivalent to the population bottleneck encountered during each daily transfer of the LTEE. Thus, regrowth maintained the representative genetic diversity of the population. Genomic DNA was isolated from several milliliters of each culture using Qiagen Genomic-tip 100/G columns. Standard DNA library preparation and sequencing on an Illumina Genome Analyzer instrument at the Michigan State University Research Technology Support Facility produced 36-bp single-end reads. These reads were mapped to the genome using the *breseq* computational pipeline (version 0.25) in polymorphism mode (*Deatherage and Barrick, 2014*). The frequencies in each sample of *gltA* mutations and other base changes characteristic of each clade were estimated by counting how many reads aligned to the position in question had the reference versus variant allele. FASTQ files have been deposited in the NCBI Sequence Read Archive (SRP051254).

## Strain construction

E. coli strains containing different *gltA*, *iclR*, and *arcB* alleles were constructed using the pKO3 allelic replacement method (*Link et al., 1997*). To create the pKO3 plasmids for allelic replacement, a 1-kb fragment including ~500 bp of flanking sequence on each side of the desired allele was PCR amplified from cells encoding the desired allele: strain REL607 (*Jeong et al., 2009*) for *gltA^wt^*; CZB154 (*Blount et al., 2012*) for *gltA1*, *iclR*, and *arcB*; and strain R1 (*Quandt et al., 2014*) for *gltA2-R*. This fragment was combined with PCR-amplified and DpnI-digested pKO3 vector backbone using the Gibson isothermal assembly method (*Gibson et al., 2009*). pKO3 plasmids containing each allele of interest were integrated into the chromosomes of recipient strains by electroporation followed by selection for chloramphenicol resistant colonies on LB agar at 43°C. Excision of the pKO3 plasmid backbone was selected for by subsequent plating on LB agar lacking NaCl and supplemented with 6% d-sucrose followed by incubation at 30°C. Sucrose resistant clones were screened for successful allelic replacement by PCR amplification of genomic DNA and Sanger sequencing.

## Growth curves

Strains were revived from frozen stocks and grown in LB. Saturated overnight cultures were diluted 1:100 into saline followed by a 1:100 dilution into the medium used for the assay and grown overnight again. Media were supplemented with kanamycin for strains containing the pCitT plasmid (*Quandt et al., 2014*). These preconditioned cultures were normalized to an $OD_{420}$ of 0.04 in saline and subsequently diluted 1:100 into the assay medium. 100 µL of each culture was then aliquoted into a 96-well flat-bottom microplate and covered with heavy mineral oil. The plate was incubated at 37°C in a Synergy HT microplate reader (Biotek; Winooski, VT). $OD_{420}$ readings were taken every 17 min with continuous shaking between readings.

## Fitness assays

Relative fitness was measured using standard LTEE co-culture competition assays (*Wiser et al., 2013*) in DM medium supplemented with carbon sources as specified. We selected spontaneous $Ara^+$ revertants of ZDB478, ZDB483, ZDB564 and ZDB706 to compete against $Ara^-$ strains (*Lenski et al., 1991*). Each of these strains was competed against its $Ara^-$ parent to verify that its fitness was unaffected by the genetic marker change (Welch's *t*-test p>0.05, $n \geq 6$).

## mRNA expression levels

DM25 flask cultures were grown to mid exponential phase ($OD_{420}$ 0.03–0.04) after preconditioning strains as described for growth curves. RNA was extracted from these cultures using the RNeasy Mini kit (Qiagen, Valencia, CA) with on-column DNase treatment. First-strand cDNA synthesis was performed from purified total RNA (0.5 µg) using the Invitrogen M-MLV reverse transcription system with a gene-specific reverse primer for *gltA* or the *ihfB* reference gene. Total cDNA (1.25 ng) and primers were added to Power SYBR Green PCR master mix (Applied Biosystems, Grand Island, NY). Quantification cycle ($C_q$) values for each reaction were determined from qPCR reactions performed on a LightCycler 96 (Roche, Indianapolis, IN). Expression levels relative to strain EQ119 were calculated from these data using the $2^{-\Delta\Delta C_q}$ method (*Livak and Schmittgen, 2001*).

## Protein purification

Citrate synthase variants were cloned into the pET-28b (+) expression vector (Novagen, Billerica, MA) to create N-terminal $His_6$ gene fusions via Gibson assembly (*Gibson et al., 2009*). The respective *gltA* gene sequences were PCR amplified from strains REL607 for *gltA^wt^*, CZB154 for *gltA1*, ZDB83 for *gltA2-7*, and ZDB107 for *gltA2-4*. Each *gltA* pET-28b (+) plasmid was transformed into E. coli strain BL21(DE3). For protein expression, cultures were grown in 500 mL LB media in 2 L Erlenmeyer flasks at 37°C with orbital shaking at 250 r.p.m. When $OD_{600}$ readings reached ~0.6, protein expression was induced with 0.5 mM IPTG and cells were incubated for 3 hr more.

For protein purification, cultures were centrifuged and cell pellets were resuspended in 6 mL of CS lysis buffer (50 mM Tris·Cl, pH 8.0; 300 mM NaCl; 20 mM imidazole). Cells were lysed by two passes through a French press. The lysate was spun down and clarified through a 0.2 µm *Supor* membrane (Pall, Port Washington, NY). Then, clarified lysate was passed through a column packed

with 1 mL Ni-NTA resin (Qiagen) that had been pre-equilibrated with CS lysis buffer. After washing with 6 column volumes of CS lysis buffer and 6 column volumes of CS wash buffer (50 mM Tris·Cl, pH 8.0; 300 mM NaCl; 50 mM imidazole), proteins were eluted in 3 mL of CS elution buffer (50 mM Tris·Cl, pH 8.0; 250 mM imidazole) and exhaustively dialyzed against CS dialysis buffer (20 mM Tris·Cl, pH 7.8; 1 mM EDTA). Protein concentrations were estimated from $A_{280}$ measurements in a Nanodrop spectrophotometer (Thermo Scientific, Waltham, MA) using an extinction coefficient of 40,715 $M^{-1}$ $cm^{-1}$ calculated by the ExPASy ProtParam tool (*Gasteiger et al., 2005*).

## Enzyme activity measurements

Citrate synthase activity of purified $His_6$-tagged GltA variants was measured using a 5′,5′-dithiobis-(2-nitrobenzoate) (DTNB) colorimetric assay (*Srere, 1969*). Readings at 412 nm were taken in 96-well plates at 25°C using a Synergy HT plate reader. Standard CS assay buffer consists of 20 mM Tris·Cl (pH 7.8), 100 mM KCl, and 1 mM EDTA (*Duckworth and Tong, 1976*). Enzyme was present at a concentration at least hundredfold lower than both substrates in all assays. Under these conditions, *E. coli* CS has been shown to conform to the ordered bisubstrate mechanism (*Anderson and Duckworth, 1988*). Kinetic data was fit to the Ordered Bi Bi equation using SigmaPlot 10 (Systat Software, San Jose, CA).

## NADH binding and inhibition measurements

NADH equilibrium binding assays were performed as previously described in CS buffer lacking KCl (*Duckworth and Tong, 1976*). Briefly, enzymes were equilibrated with varying concentrations of NADH (0–6.4 μM) for 1 hr at 25°C. Fluorescent measurements were made with excitation at 340 nm and emission reading at 425 nm in a M200 plate reader (Tecan, Männedorf, CHE). The observed changes in fluorescence versus NADH concentration were fit to a hyperbolic ligand-binding curve using SigmaPlot 10.

Inhibition assays were performed essentially as described elsewhere (*Stokell et al., 2003*). Varying concentrations of NADH were equilibrated with enzyme in CS buffer lacking KCl at 25°C for 1 hr. Substrates OAA and AcCoA were both added at 100 μM and initial reaction rates were measured using the DTNB assay described above. Enzyme activities were normalized to the activity of wild-type GltA in the absence of NADH. Percent inhibition was plotted versus NADH concentration and fit to a hyperbolic model in SigmaPlot 10.

## Molecular modeling

Wild-type and mutant citrate synthase structures were analyzed using the Molecular Operating Environment (MOE2013.08). Dimeric models of ligand-free and NADH-bound *E. coli* citrate synthase (PDB: 1NXE and 1NXG) (*Maurus et al., 2003*) were prepared for analysis by reverting alanine-383 to phenylalanine and processing with the Structure Preparation application within MOE. Each model was then protonated (37°C, pH 7.4, 0.1 salt) using Protonate3D. As further preparation for QM/MM analyses, non-bridging solvent molecules were removed, a 6 Å solvent sphere was added, and charges were neutralized by the addition of KCl. Lastly, amino acid substitutions in a given GltA variant were added sequentially with energy minimization to an RMS gradient of $10^{-3}$ kcal/mol/$Å^2$ using the Amber12 force field with Extended Hückel Theory and R-field solvation electrostatics after each mutation. NADH binding energies for the resulting models were calculated using the Ligand Interactions subroutine.

## Flux balance analysis

Flux balance analysis was performed with the COBRA Toolbox v2.0 (*Schellenberger et al., 2011*) and MATLAB v7.10.0 (The MathWorks, Inc., Natick, MA) using the glpk solver. The genome-scale model of *E. coli* strain REL606 metabolism, iECB_1328 (*Monk et al., 2013*), was used for this analysis. The model incorporates 2,750 reactions and 1954 metabolites. Default media conditions and reaction bounds were used. Carbon source uptake fluxes were set at 10 mmol per gram dry cell weight per hour unless otherwise stated. To simulate combined utilization of citrate and glucose, the uptake fluxes for each carbon source were adjusted to match the molar ratio at which they are present in DM25 (10:1). We used flux variability analysis to predict the full ranges of flux values for reactions in central metabolism that exist within the set of optimal global solutions. There was no

variability in the optimal flux predicted for the citrate synthase reaction under any of the conditions we tested.

## Acknowledgements

We thank Richard Lenski, Dacia Leon, Christopher Marx, Noah Ribeck, Caroline Turner, and Johnny Blazeck for helpful discussions. We acknowledge the use of resources at the Texas Advanced Computing Center (TACC). This work was supported by grants from the US National Institutes of Health (R00-GM087550 to JEB), the US National Science Foundation BEACON Center for the Study of Evolution in Action (DBI-0939454 to JEB. and ZDB), the US Army Research Office (W911NF-12-1-0390 to JEB), the US Defense Advanced Research Projects Agency (to GG and ADE), the US Defense Threat Reduction Agency (to GG and ADE), and the John Templeton Foundation (RFP 12-13 to ZDB).

## Additional information

### Funding

| Funder | Grant reference number | Author |
| --- | --- | --- |
| National Science Foundation | DBI-0939454 | Zachary D Blount<br>Jeffrey E Barrick |
| Army Research Office | W911NF-12-1-0390 | Jeffrey E Barrick |
| Defense Advanced Research Projects Agency | | Andrew D Ellington<br>George Georgiou |
| National Institutes of Health | R00-GM087550 | Jeffrey E Barrick |
| John Templeton Foundation | RFP 12-13 | Zachary D Blount |
| Defense Threat Reduction Agency | | Andrew D Ellington<br>George Georgiou |

The funders had no role in study design, data collection and interpretation, or the decision to submit the work for publication.

### Author contributions

EMQ, JG, ZDB, Conception and design, Acquisition of data, Analysis and interpretation of data, Drafting or revising the article; ADE, GG, JEB, Conception and design, Analysis and interpretation of data, Drafting or revising the article

### Author ORCIDs

Erik M Quandt, http://orcid.org/0000-0002-3287-7777
Jeffrey E Barrick, http://orcid.org/0000-0003-0888-7358

## Additional files

### Supplementary files

• Supplementary file 1. *E. coli* strains and plasmids used in this study.

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
