## [Decision Letter]

Thank you for submitting your work entitled "Fine-tuning citrate synthase flux potentiates and refines metabolic innovation in the Lenski evolution experiment" for peer review at *eLife*. Your submission has been favorably evaluated by Ian Baldwin (Senior Editor) and three reviewers, one of whom, Daniel Kliebenstein, is a member of our Board of Reviewing Editors.

The reviewers have discussed the reviews with one another and the Reviewing editor has drafted this decision to help you prepare a revised submission).

The paper looks at the evolution of primary metabolism within a long-term evolution experiment and shows that this occurs via a non-linear process. All reviewers felt that the work was very nicely conducted and will be highly useful to researchers studying evolution or metabolism.

Essential revisions:

It was felt that the figures needed to be significantly clarified to improve the ability of the reader to derive the messages. This is not meant to decrease the information in each figure but to make the presentation more direct. Given that *eLife* has no figure or length limitations it was felt that this was imminently possible and would greatly improve the accessibility of the manuscript. Aside this revision, please take into consideration the suggestions made by the reviewers.

Reviewer #1:

This manuscript uses a long-term evolution experiment to explore the ability of central metabolism to evolve from one steady state network to a second steady state network. This shows that even in central metabolism that the path of evolution is not a straight forward walk but involves different paths in specific sequences. Extending this to primary metabolism will help to shift our current thinking on how "conserved" primary metabolism is.

In Figure 1, it was difficult to interpret the ancestry of the secondary *gltA* alleles. This figure would probably be served better if part A showed haplotypes rather than individual alleles and the haplotypes were used for the coloring in part B. It would also be helpful if these colors were maintained throughout the figures to allow the reader to follow the data.

In all figures, evidence of statistical support for claims of difference needs to be provided. SE bars are not the equivalent of statistical support. 95% CI are more acceptable for this purpose.

Reviewer #2:

Quandt et al. investigate the mutational pathway that allowed the evolution of citrate consumption in one population of *E. coli* in the Lenski Long-Term Experiment. It was known that a mutation involving *citT* was important for initial citrate uptake and a second mutation in *dctA* was essential for full utilization of the resource. However, the rarity of the adaptation (occurring in only 1 of 12 populations) suggested that other potentiating mutations were involved. The authors argue that a mutation in citrate synthase (*gltA*) was a critical for enabling citrate consumption to evolve. They show that the *gltA1* mutation arose before the *E. coli* evolved citrate import, and that the initial *citT* mutation is deleterious in the absence of *gltA1*. The authors demonstrate that *gltA1* can increase fitness on acetate, but that the mutation becomes deleterious once cells have acquired the ability to utilize citrate. A second set of mutations (termed *gltA2*) is able to compensate for the cost of *gltA1*. The authors show that *gltA1* increases activity of citrate synthase by removing allosteric inhibition by NADH, while subsequent mutations in the gene reduce activity of the protein through a variety of methods. Finally Quandt et al. use metabolic modeling to show that optimal growth on acetate involves increased flux through citrate synthase relative to glucose, and that optimal growth on citrate requires no flux through this pathway. Combining these bits of evidence the authors suggest that utilization of citrate involved a mutation in *gltA* that improved growth on acetate and potentiated the *citT* mutation, by increasing flux through TCA. Once cells became strong citrate utilizers, the initial *gltA* mutation became deleterious driving the evolution of a diversity of secondary mutations to compensate for its cost. This manuscript represents an extensive amount of work, and I think it will be of great interest to many in the experimental evolution community.

I would like to see a little more attention paid to the significance of the research. I appreciate that the authors do not oversell their work, but I think the manuscript would greatly benefit from a few more sentences about how this relates to our general understanding of evolutionary processes. Additionally, it seems like some of the results do not add directly to the narrative that is being developed.

*Reviewer #3:*

This is a lovely piece of work. It fills in so many gaps and delivers something of real substance. For the most part I found it a pleasure to read and digest.

The authors could do far better with their presentation of data in Figure 5 and Figure 6. To me, as someone who knows the TCA cycle and even cares about it, I struggled to make sense of Figure 5. Figure 6 similar left me confused. I appreciate the attempt to present the full picture – and this absolutely must be retained – but the reader has to be taken through the data. At the present time there is simply too much information. This frustrates me because I want to understand the full weight of the data. After pouring over Figure 5 and Figure 6, the caption and the text, I began to wonder whether I had fully understood what had come before.

---

## [Author Response]

*Essential revisions:*

*It was felt that the figures needed to be significantly clarified to improve the ability of the reader to derive the messages. This is not meant to decrease the information in each figure but to make the presentation more direct. Given that* eLife *has no figure or length limitations it was felt that this was imminently possible and would greatly improve the accessibility of the manuscript.*

We have sought to clarify the information in the figures by splitting the original 6 figures into 9 figures and by greatly expanding upon the context provided in several figure captions so that they walk the reader through the data/models and their significance.Reviewer #1: *This manuscript uses a long-term evolution experiment to explore the ability of central metabolism to evolve from one steady state network to a second steady state network. This shows that even in central metabolism that the path of evolution is not a straight forward walk but involves different paths in specific sequences. Extending this to primary metabolism will help to shift our current thinking on how "conserved" primary metabolism is. In Figure 1, it was difficult to interpret the ancestry of the secondary gltA alleles. This figure would probably be served better if part A showed haplotypes rather than individual alleles and the haplotypes were used for the coloring in part B. It would also be helpful if these colors were maintained throughout the figures to allow the reader to follow the data.*

Since there is no recombination in this *E. coli* evolution experiment (the bacteria are strictly asexual), haplotype and allele frequencies are equivalent. The "Muller plot" in Figure 1 is a standard way of illustrating genetic dynamics in an asexual population, with sectors representing new mutations nested within previous ones to show linkage.

Since it may not be perfectly clear from the figure that all of the *gltA1* alleles arise separately (in independent lineages derived the ancestral genotype with the *gltA* allele, and not nested within one another), we have added a statement to clarify this point to the figure legend: "Each of the *gltA2* alleles (shaded as in panel A) evolved independently and defines a separate evolved genotype and all of its descendants."

We think that keeping the colors of the *gltA2* alleles all as similar shades of blue/purple (rather than representing them with entirely distinct colors) is the most useful way of showing our the data. It emphasizes that all *gltA2* alleles seem to have similar affects on cellular metabolism (decreasing citrate synthase flux). So, one can rapidly assess the overall total frequency of cells with any *gltA2* allele in the LTEE population over time in Figure 1 by following all blue/purple sectors together as one visual category. Depicting the *gltA2* alleles all in a similar shade also makes it easier to illustrate interactions (in the remaining figures) between a generic *gltA2* allele and other key evolved alleles (*gltA1, citT, dctA*, iclR*, etc.) that we do represent using very distinct colors.

In any case, we always label the exact *gltA2* allele that is being studied and do currently maintain the same (subtle) color scheme for distinguishing among them in other figures.

*In all figures, evidence of statistical support for claims of difference needs to be provided. SE bars are not the equivalent of statistical support. 95% CI are more acceptable for this purpose.*

Thank you for this comment. We have altered the gene expression data to show 95% CI instead of SE bars since we rely on this figure to show statistical significance (Figure 5). The fitness measurements already use 95% confidence intervals throughout. All of the arguments that we make related to growth curve data are based on very large differences, for which there is really no question that they are statistically significant.

Reviewer #2:*Quandt et al. investigate the mutational pathway that allowed the evolution of citrate consumption in one population of* E. coli *in the Lenski Long-Term Experiment. It was known that a mutation involving* citT *was important for initial citrate uptake and a second mutation in* dctA *was essential for full utilization of the resource. However, the rarity of the adaptation (occurring in only 1 of 12 populations) suggested that other potentiating mutations were involved. The authors argue that a mutation in citrate synthase (*gltA*) was a critical for enabling citrate consumption to evolve. They show that the* gltA1 *mutation arose before the* E. coli *evolved citrate import, and that the initial* citT *mutation is deleterious in the absence of* gltA1*. The authors demonstrate that* gltA1 *can increase fitness on acetate, but that the mutation becomes deleterious once cells have acquired the ability to utilize citrate. A second set of mutations (termed* gltA2*) is able to compensate for the cost of* gltA1*. The authors show that* gltA1 *increases activity of citrate synthase by removing allosteric inhibition by NADH, while subsequent mutations in the gene reduce activity of the protein through a variety of methods. Finally Quandt et al. use metabolic modeling to show that optimal growth on acetate involves increased flux through citrate synthase relative to glucose, and that optimal growth on citrate requires no flux through this pathway. Combining these bits of evidence the authors suggest that utilization of citrate involved a mutation in* gltA *that improved growth on acetate and potentiated the* citT *mutation, by increasing flux through TCA. Once cells became strong citrate utilizers, the initial* gltA *mutation became deleterious driving the evolution of a diversity of secondary mutations to compensate for its cost. This manuscript represents an extensive amount of work, and I think it will be of great interest to many in the experimental evolution community.*

*I would like to see a little more attention paid to the significance of the research. I appreciate that the authors do not oversell their work, but I think the manuscript would greatly benefit from a few more sentences about how this relates to our general understanding of evolutionary processes. Additionally, it seems like some of the results do not add directly to the narrative that is being developed.*

We have added a paragraph to the end of the paper to further speak to the broader significance of our findings:

“More broadly, our results demonstrate that evolutionary innovations may rely not only on the acquisition of novel genes or the co-option of molecular machinery […] that lead to the successful colonization of new niches.”

Reviewer #3:*This is a lovely piece of work. It fills in so many gaps and delivers something of real substance. For the most part I found it a pleasure to read and digest. The authors could do far better with their presentation of data in Figure 5 and Figure 6. To me, as someone who knows the TCA cycle and even cares about it, I struggled to make sense of Figure 5. Figure 6 similar left me confused. I appreciate the attempt to present the full picture – and this absolutely must be retained – but the reader has to be taken through the data. At the present time there is simply too much information. This frustrates me because I want to understand the full weight of the data. After pouring over Figure 5 and Figure 6, the caption and the text, I began to wonder whether I had fully understood what had come before.*

We have split each of these figures into two new figures and greatly expanded the figure captions in hopes of better explaining the data being presented and putting it in context.